## Report

# Partially inserted nascent chain unzips the lateral gate of the Sec translocon

Lukas Kater[1], Benedikt Frieg[2] iD, Otto Berninghausen[1], Holger Gohlke[2,3,*] iD, Roland Beckmann[1,**] iD & Alexej Kedrov[1,4,***] iD

## Abstract

The Sec translocon provides the lipid bilayer entry for ribosome-bound nascent chains and thus facilitates membrane protein biogenesis. Despite the appreciated role of the native environment in the translocon:ribosome assembly, structural information on the complex in the lipid membrane is scarce. Here, we present a cryo-electron microscopy-based structure of bacterial translocon SecYEG in lipid nanodiscs and elucidate an early intermediate state upon insertion of the FtsQ anchor domain. Insertion of the short nascent chain causes initial displacements within the lateral gate of the translocon, where α-helices 2b, 7, and 8 tilt within the membrane core to "unzip" the gate at the cytoplasmic side. Molecular dynamics simulations demonstrate that the conformational change is reversed in the absence of the ribosome, and suggest that the accessory α-helices of SecE subunit modulate the lateral gate conformation. Site-specific cross-linking validates that the FtsQ nascent chain passes the lateral gate upon insertion. The structure and the biochemical data suggest that the partially inserted nascent chain remains highly flexible until it acquires the transmembrane topology.

**Keywords** membrane protein insertion; nanodisc; native environment; reconstitution; ribosome

**Subject Categories** Membranes & Trafficking; Structural Biology

See also: **Y Tanaka & T Tsukazaki et al** (October 2019)

## Introduction

Membrane proteins constitute a large part of the cellular proteome and determine the vital functionality and identity of biological membranes. These proteins are co-translationally targeted as ribosome: nascent chain complexes (RNCs) to the endoplasmic reticulum in eukaryotes and the cytoplasmic membrane in bacteria and archaea, where they are inserted by the dedicated and universally conserved Sec translocon (Fig 1A and B) [1]. The translocon, an integral membrane protein itself, builds a protein-conducting channel in the lipid bilayer and allows either transmembrane passage of nascent polypeptide chains or their partitioning into the lipid environment as transmembrane α-helices (TMHs). The nascent chain hydrophobicity forms a basis for the triage [2]. The central subunit of the translocon, SecY in bacteria or Sec61α in eukaryotes, consists of 10 TMHs arranged as a pseudo-symmetric "clam-shell" with a protein-conducting pore between the N- and C-terminal parts (Fig 1) [3,4]. A bilayer-facing crevice between SecY TMHs 2b and 7 is assumed to serve as a route, or a "lateral gate", for nascent TMHs to reach the hydrophobic membrane core. SecY is stabilized at the periphery by the essential subunit SecE/Sec61γ that contains two α-helices, one in interfacial and one in transmembrane topologies. SecE of some Gram-negative bacteria, including *Escherichia coli*, contains also an accessory pair of N-terminal TMHs, the role and localization of which have remained largely unclear [5]. A non-essential and non-conserved SecG/Secβ subunit near the N-terminal half of SecY is built of either one or two TMHs and plays a stimulatory role in protein translocation [6].

The assembly of the translocon:ribosome complex at the cytoplasmic membrane interface is a key step in membrane protein biogenesis, as it allows the hydrophobic nascent chain to egress into the lipid bilayer via the translocon, while not being exposed to the polar aqueous environment [1,7]. The architecture of the complex has been extensively studied by structural methods, first of all cryo-electron microscopy (cryo-EM) [8–11]. Binding of a ribosome results in minor rearrangements within the translocon and brings it to a pre-open or "primed" state [11]. The following insertion of a sufficiently hydrophobic helical domain, such as a signal sequence or signal anchor domain, shifts the complete N-terminal domain of SecY/Sec61α by 22° and also tilts TMH 7, so the lateral gate of the translocon acquires an open state (Fig 1B) [12,13]. The folded signal

1 Gene Center Munich, Ludwig-Maximilian-University, Munich, Germany
2 John von Neumann Institute for Computing, Jülich Supercomputing Centre, Institute for Complex Systems - Structural Biochemistry (ICS-6), Forschungszentrum Jülich GmbH, Jülich, Germany
3 Institute for Pharmaceutical and Medicinal Chemistry, Heinrich Heine University Düsseldorf, Düsseldorf, Germany
4 Synthetic Membrane Systems, Institute for Biochemistry, Heinrich Heine University Düsseldorf, Düsseldorf, Germany
*Corresponding author. Tel: +49 211 81 13662; E-mail: gohlke@hhu.de
**Corresponding author. Tel: +40 89 2180 76900; E-mail: beckmann@genzentrum.lmu.de
***Corresponding author. Tel: +49 211 81 13731; E-mail: kedrov@hhu.de

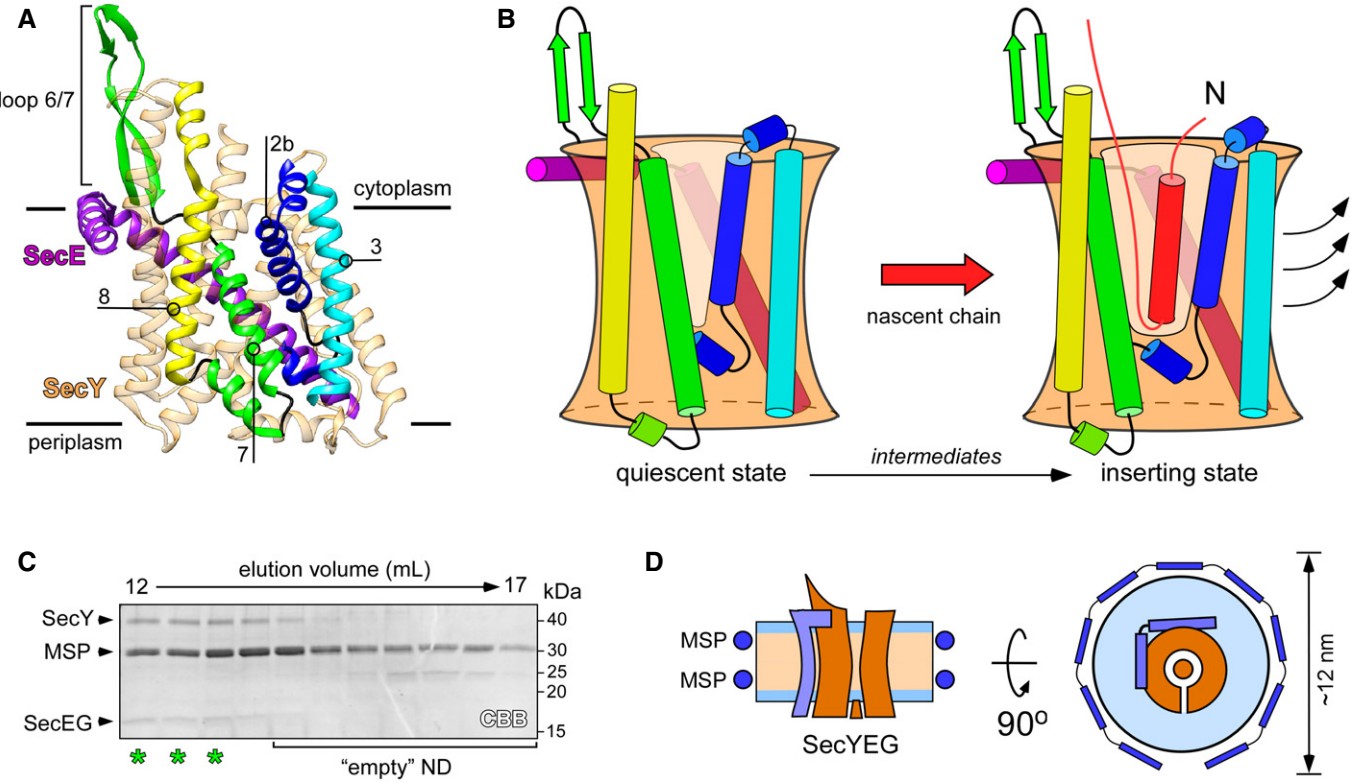

**Figure 1. Structure and dynamics of SecYEG translocon.**

A   Structure of quiescent SecYEG of *Thermus thermophilus* in the lipid cubic phase (PDB ID: 5AWW). TMHs 2b, 3, 7, and 8 of the lateral gate, as well as the proximate loop 6/7 involved in ribosome binding are indicated. The non-essential SecG subunit is omitted for clarity.

B   Model of the SecY lateral gate opening upon inserting a nascent chain (red) in the lipid bilayer. The color-coding of SecYE TMHs is as in panel (A). In the presence of the completely inserted and folded nascent chain, TMHs 2b and 3 of the N-terminal domain of SecY are displaced (arrows) thus opening a broad passage for the nascent TMH toward the lipid moiety.

C   SDS–PAGE of SecYEG-ND sample after size-exclusion chromatography. Asterisks indicate translocon-enriched fractions used for forming the RNC FtsQ:SecYEG-ND complex. Lipid-loaded "empty" nanodiscs elute at larger volumes and so can be separated.

D   Schematic drawing of a SecYEG-ND particle. Lateral dimensions of the nanodisc should be appropriate to accommodate a single SecYEG with surroundings lipids, thus mimicking the naturally occurring environment.

sequence in a transbilayer topology may occupy the lateral gate where it replaces TMH 2b. Upon the further elongation of the nascent polypeptide chain, the newly inserted α-helix leaves the lateral gate and egresses into the lipid bilayer, and the translocon undergoes a reverse transition from a widely opened [14] to a compact, pre-closed state [15].

Although the dynamics of the lateral gate have been commonly acknowledged [16,17], the mechanism of the nascent chain insertion remains unclear. First, existing structures reflect rather late insertion stages, where the signal sequence has been fully inserted in the transmembrane topology, while early intermediates have been barely addressed [4,18]. Second, a vast majority of available ribosome:translocon structures represent detergent-solubilized complexes; however, the non-physiological environment and extensive downstream purification schemes may significantly affect the conformation and the interaction properties of membrane proteins, including the translocon [19–21]. The variations in detergent-based solubilization protocols may explain contradictory results on the translocon dynamics, where either a local displacement of helices within the lateral gate or an extensive movement of the complete

N-terminal half was observed upon the nascent chain insertion, and also the conformation of the central "plug" domain has been disputed [12,13,22]. Furthermore, a compact "primed" state has been described for detergent-solubilized translocons in the absence of hydrophobic nascent chains [11], while a recent cryo-electron tomography analysis has revealed a predominantly open conformation of the ribosome-bound Sec61 within native ER membranes and so suggested a crucial effect of the molecular environment on protein dynamics [23].

Up to date, the only structure of the translocon:ribosome complex at the lipid interface was obtained by cryo-EM when using nanodisc-reconstituted SecYEG (SecYEG-ND) bound to a translation-stalled RNC [14]. Although demonstrating an advance compared to detergent-solubilized systems, the structure offers only limited resolution and also illustrates a rather late stage of the TMH insertion, with the translocon lateral gate widely open and the inserted anchor domain de-localized within the membrane. Here, we set out to determine the structure of the SecYEG:RNC complex that would describe an early stage of a transmembrane domain insertion into the lipid bilayer. Using cryo-EM and single-particle

analysis, we resolved for the first time all three subunits of SecYEG in nanodiscs and described a novel conformation, where SecY TMHs 2b and 7 were apart at the cytoplasmic side to form a V-shaped lateral gate that is pre-opened for the nascent chain insertion, while accessory SecE TMHs 1 and 2 interacted with the gate at the periplasmic side. The RNC-induced dynamics within the translocon was validated by atomistic molecular dynamics simulations, which also described the interactions of SecYEG with anionic lipids. Cryo-EM data and site-specific chemical cross-linking further suggested that the FtsQ anchor domain is inserted via the lateral gate, where it forms close contacts with SecY TMH 7, but remains highly flexible before leaving the translocon.

## Results and Discussion

Functional reconstitution of *E. coli* SecYEG in nanodiscs has been previously performed by several groups for biochemical, biophysical, and structural studies and allowed probing of the translocon interactions with the motor protein SecA, targeting factors, and ribosomes [14,20,24,25]. The diameter of formed nanodiscs is essentially determined by the length of the major scaffold protein (MSP) that girdles the lipid bilayer [26,27]. Translocon molecules have been initially embedded into nanodiscs as small as 9 nm in diameter [16,20,24]. However, a follow-up functional analysis demonstrated that larger nanodisc dimensions are beneficial for facilitating the translocation activity, likely due to the increased amount of co-reconstituted lipids [25,28]. Thus, we used an extended scaffold protein MSP1E3D1 and POPG/POPC lipids to reconstitute SecYEG into nanodiscs with a diameter of approximately 12 nm. A large excess of MSPs and lipids ensured that translocons were reconstituted predominantly as monomers [25], as those have been shown to be the principle functional form both in bacteria and in eukaryotes [9,29,30]. Due to solvent-exposed loops of SecYEG, which contributed to the hydrodynamic radius, SecYEG-ND could be separated from "empty" nanodiscs containing only lipids by means of size-exclusion chromatography (Fig 1C). Within formed nanodiscs, SecYEG would occupy ~30% of the surface area (Fig 1D) [25,26,28], thus providing sufficient space for the conformational dynamics, and for insertion of nascent TMHs upon interactions with RNCs.

We have previously demonstrated that SecYEG:ribosome assembly is strongly enhanced by hydrophobic nascent chains, such as a TMH of FtsQ, a model protein for studying the SecYEG-mediated insertion pathway [20]. The hydrophobic polypeptide exposed from a ribosome exit tunnel is sufficient to mediate SecYEG:ribosome binding in native and model membranes, even in the absence of targeting factors [20,31], but unlikely to undergo the complete insertion due to its short ribosome-bound linker. Thus, to investigate an early stage of the TMH insertion, we prepared translation-stalled ribosomes, which exposed the first 48 amino acids of FtsQ, including the TMH within the nascent chain (Fig EV1), and incubated those with a 10-fold excess of SecYEG-ND to achieve complex formation. After vitrification, samples were subjected to cryo-EM imaging and single-particle analysis. RNCs could be readily seen in raw micrographs, and a discoidal density of SecYEG-ND bound to RNCs was observed in projection groups of two-dimensional (2D) classification and in 3D reconstructions (Fig 2A–C). After sorting and refinement steps (Fig EV2), the ribosome structure was resolved at 3.3 Å, and

independent refinement of the SecYEG-ND:RNC complex elements led to 3.2 and 3.1 Å resolution for the small (30S) and large (50S) ribosomal subunits, respectively (Appendix Fig S1), and was limited to 6 Å for the lipid-embedded SecYEG due to its small size and apparent dynamics relative to the 50S ribosomal subunit (Movie EV1). The local resolution within the SecYEG-ND particle ranged from 3.5 Å at the ribosome contact sites to 6–7 Å within the transmembrane core and above 10 Å for the surrounding MSP1E3D1 and lipid head groups, which could be visualized at lower threshold levels (Fig 2D and E).

In agreement with the initial prediction, the nanodisc dimensions were sufficiently large to accommodate a single copy of SecYEG. As SecYEG was positioned in the center of the nanodisc and contacts with edges of the lipid bilayer or MSP were not observed, it is likely that the translocon conformation was not affected by the confined environment. As electron densities of the centrally positioned translocon and the MSP were well-separated (Fig 2E), it facilitated the assignment of rod-shaped densities to TMHs of SecYEG and building the molecular model based on the structure of the quiescent translocon [4]. Both TMHs and extramembrane domains of SecY, SecE, and SecG subunits could be unambiguously fitted into the cryo-EM density (Figs 2E and 3A). The translocon:ribosome complex was established via the well-known canonical interactions [9,11,14]: Two structured cytoplasmic loops between TMHs 6/7 and 8/9 of SecY extended toward the ribosomal tunnel to interact with rRNA helices H6, H24, and H50, and the uL23 protein. Additionally, the ribosomal protein uL24 approached the C-terminal end of the SecY TMH 10, and the ribosomal protein uL23 formed two contacts within the essential amphipathic helix of SecE. Differently to earlier findings [14], we did not observe the contact between the rRNA helix H59 and the lipid head groups, although the H59 helix was displaced toward the bilayer (Fig 3B). It seems plausible that those contacts are established at a later stage of membrane protein insertion, when one or more nascent TMHs egress the lipid bilayer and the H59 helix "screens" the charge of connecting loops, and so participates in the topology determination [15,32]. When evaluating other known structures of bacterial and eukaryotic translocons in complex with ribosomes (Appendix Fig S2), we noted a close agreement between our model and the detergent-solubilized *E. coli* SecYEG bound to a translation-stalled ribosome [18]. Interestingly, although the SecYEG structures in both environments were highly similar, the relative orientation of the ribosome and SecYEG differed substantially: While being bound to the RNC via its C-terminal domain, the detergent-solubilized translocon rotated as a rigid body away from the rRNA helix H59, so the displacement was most pronounced for its N-terminal half (Fig EV3). It is tempting to speculate that the altered SecYEG:ribosome binding geometry, as well as the enhanced affinity of the complex in detergent [20], arose from the lack of electrostatic interactions between the rRNA and the polar moiety of lipid head groups.

In spite of the loose binding of SecYEG to the RNC and its higher flexibility, the complete architecture of essential SecY and SecE subunits was resolved, and a single-helix density proximate to the SecY N-terminal domain was assigned to TMH 2 of the SecG subunit, while TMH 1 could not be reliably detected (Figs 2E and 3A). No SecG subunit could be resolved in the earlier structure of SecYEG-ND [14], and the crystal structure of the quiescent SecYEG revealed that SecG TMH 1 faces away from the translocon core, so its periplasmic tip is separated by ~10 Å from the nearest TMH 4 of SecY, with a

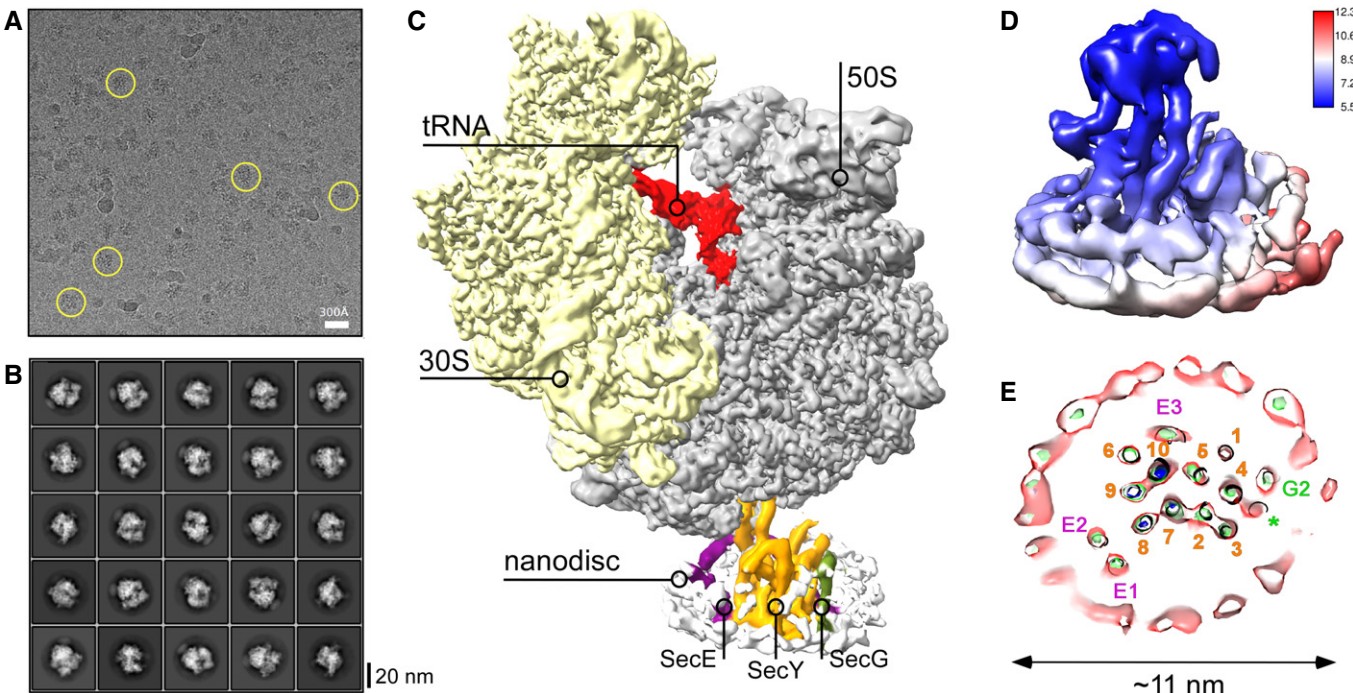

**Figure 2. Cryo-EM of the RNC FtsQ:SecYEG-ND complex.**

A   Representative cryo-EM micrograph of RNC FtsQ:SecYEG-ND. Exemplary individual ribosomes are encircled.

B   Examples of two-dimensional classes of imaged particles. RNC:nanodisc assemblies can be seen at different view angles.

C   Three-dimensional reconstruction of RNC FtsQ:SecYEG-ND complex. Primary structural elements of the ribosome and SecYEG-ND are indicated.

D   Local resolution map of SecYEG-ND sub-particle. The cytoplasmic side of the translocon demonstrates higher resolution due to stabilization by the bound ribosome, while high resolution at the periplasmic side is hindered by the SecYEG-ND dynamics within the complex. The associated ribosome is not shown for clarity.

E   A planar slice through the SecYEG-ND core at different signal levels (blue/green/red) with indicated positions of SecYEG TMHs (SecY indicated in orange, SecE in purple, and SecG in green). A single helical turn could be fitted in a density in the area where SecG TMH 1 was expected (green asterisk).

lipid molecule filling the void [4]. Thus, weak protein:protein inter-subunit interactions in the lipid environment likely favor spatial dynamics of SecG, up to a complete topology inversion [33], and the dynamics might be modulated by the ribosome binding. Remarkably, within the SecYEG-ND complex we could clearly observe accessory TMHs 1 and 2 of SecE, which were either absent or only poorly resolved in previous translocon structures [14,15,18]. Earlier models placed the SecE TMHs either distanced from the translocon by 20 Å, or near SecY TMH 9, i.e., at the back of the translocon [14,15] (Appendix Fig S2). However, our structure revealed a very different organization of the complex, as SecE TMHs formed a helical hairpin in close proximity to SecY C-terminal domain, and the hairpin was tilted within the lipid bilayer by ~30° (Fig 3A). Such a tilted orientation of the SecE TMHs could also be recognized in densely packed 2D crystals of SecYEG [34,35], but has not been reported for either free-standing or ribosome-bound translocons. Surprisingly, the periplasmic loop of the SecE helical hairpin reached TMH 8 and a short helix connecting TMHs 7 and 8 of SecY, and so appeared in direct contact with the lateral gate of the translocon, thus suggesting a potential role of SecE in the translocon gating mechanism but also explaining interactions of SecE with nascent TMHs soon after their membrane partitioning [36].

We further examined whether the early interactions with the RNC were sufficient to trigger a conformational change within

SecYEG, as it would be required for the nascent chain insertion into the lipid bilayer. SecY TMH 2a, known as a plug domain [37,38], resided in the central position, thus keeping the SecY pore sealed upon RNC binding [12,31], and only minor shifts could be seen for most TMHs in comparison with the quiescent state or detergent-solubilized SecYEG:RNC complex [4,18] (Fig 3C and Appendix Fig S2). Interestingly though, substantial rearrangements were observed within the lateral gate of the translocon, when compared both to the quiescent and to RNC-bound detergent-solubilized states (Fig 3D): TMH 2b was displaced toward the central pore of the translocon, and SecY TMH 7 underwent a tilting of ~ 5°, so its cytoplasmic and the periplasmic ends approached TMH 8 and TMH 3, respectively [3,4]. This tilting of TMH 7 was coupled to a displacement of TMH 8, as they are connected via a short rigid helix at the periplasmic side (Fig 3D). The resulting conformation of the ribosome-bound translocon manifested a V-shaped crevice at the cytoplasmic side of the lateral gate that differed from the rather closed conformation of the detergent-solubilized SecYEG [18], but also from "primed" and fully opened post-insertion states of the eukaryotic homolog [10,11,13]. Thus, the observed conformation likely reflected a novel early stage in the gate opening. Such dynamics are in agreement with a previous fluorescence-based study on SecYEG-ND:RNC [16], but, to our knowledge, represent the first direct visualization of the pre-opened translocon in the lipid environment.

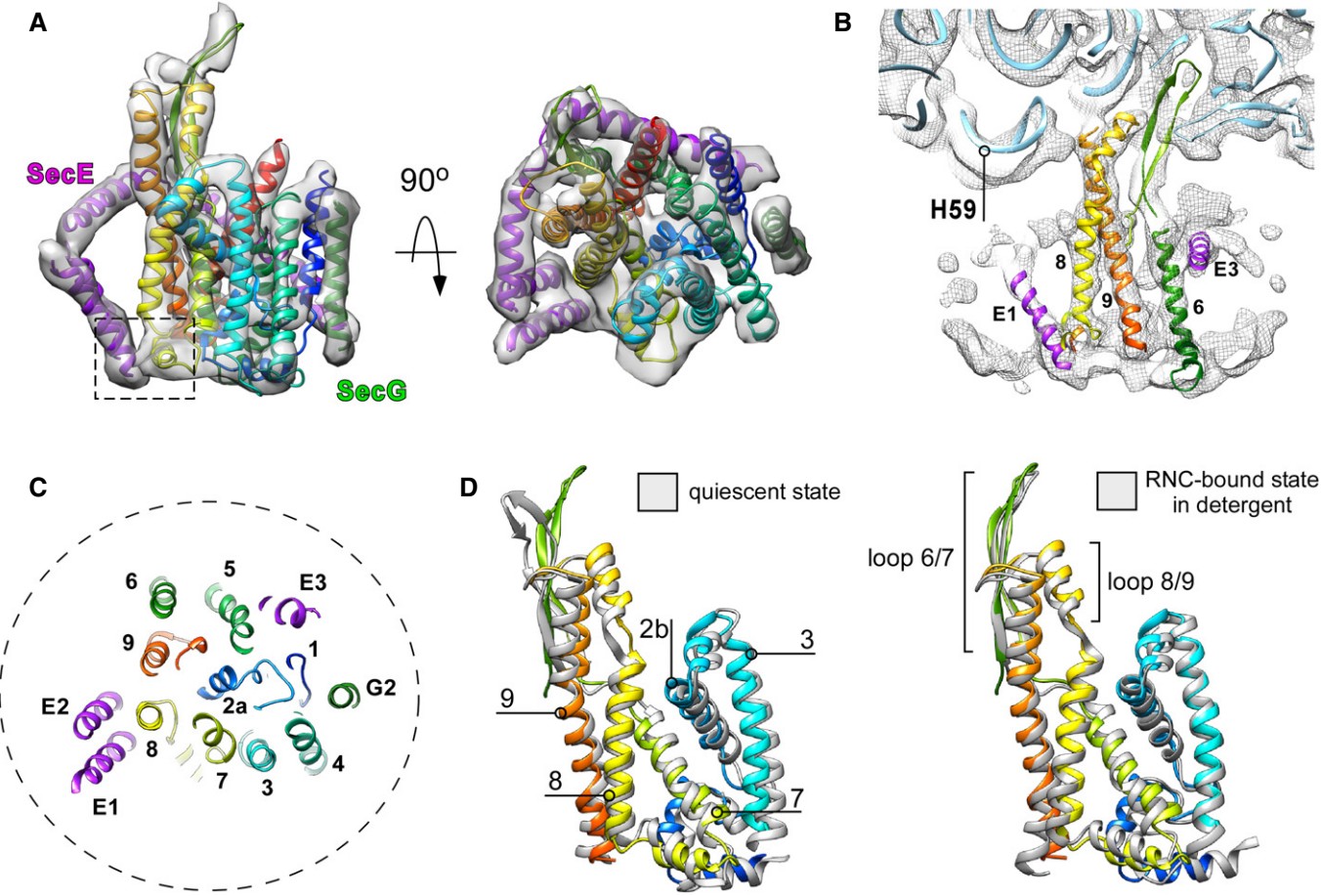

**Figure 3. Structural dynamics of the translocon and ribosome upon the nascent chain insertion.**

A   Isolated cryo-EM density of SecYEG with the fitted molecular model of the translocon in front and the cytoplasmic views. SecY is displayed in rainbow pattern, SecE in purple, and SecG in green. Dashed box: a contact site between tilted SecE TMHs 1/2 and SecY TMH 8.

B   Cryo-EM density corresponding to the ribosomal RNA helix 59 ("H59") is displaced toward the nanodisc. No contact with the lipid bilayer could be detected. Blue ribbon: structure of the translocon-free 50S ribosomal subunit (PDB ID: 4UY8).

C   Central cross-section through the SecYEG model. The "plug" TMH 2a occupies the central position, thus keeping the translocon sealed. The nanodisc perimeter is indicated as a dashed circle.

D   The lateral gate of nanodisc-embedded translocon undergoes rearrangements relatively to a quiescent conformation (left, PDB ID 5AWW) and an RNC-bound detergent-solubilized state (right, PDB ID 5GAE).

To investigate whether the observed translocon conformation was a result of RNC FtsQ binding, we employed microsecond-long molecular dynamics (MD) simulations of SecYEG in explicit solvent and an explicit membrane, which allows to study the behavior of lipid-embedded SecYEG in full atomic detail [39]. From the projection of MD conformations of SecY onto the plane spanned by the first two principal components (PC; both PCs together describe ~50% of the total variance of motions during the simulations), a configurational free energy landscape was computed (equation 1). In this landscape, the SecY conformation from the SecYEG-ND:RNC complex lies in an area of slightly elevated free energy ($\Delta G_{conf.,i} \approx 2$ kcal/mol, Appendix Fig S3A), suggesting that this conformation was stabilized by the bound RNC and/or the nascent chain. The mechanism of structural adaptation of the translocon was then probed in a reverse direction, as the MD simulations started from the RNC-bound SecYEG conformation, but without

RNC FtsQ. That way, the adaptation toward a non-disturbed quiescent state could be followed, as has previously been shown for membrane protein complexes [40,41]. The cytoplasmic loop 6/7 of SecY was highly mobile (mean root-mean-square fluctuations (RMSF) > 5 Å; Fig 4A), likely due to the absent ribosome that otherwise recruits the loop as a docking site. The TMHs were substantially less dynamic (RMSF < 3 Å), except for the lateral gate and the cytoplasmic part of TMH 2b. Structural differences upon reaching the free energy minimum were the most substantial for loop 6/7 and were followed by the lateral gate (Fig 4B). We measured internal distances within the lateral gate (TMHs 2, 7, and 8), between TMH 7 and the adjacent TMH 3, as well as the angle $\eta$ between TMH 7 and TMH 8 (Fig EV4, panels A and C). The cryo-EM structure implied that binding of RNC FtsQ to SecY induced tilting of TMH 7, such that its periplasmic end approached TMH 3, while TMH 2b shifted toward the pore. This effect was completely

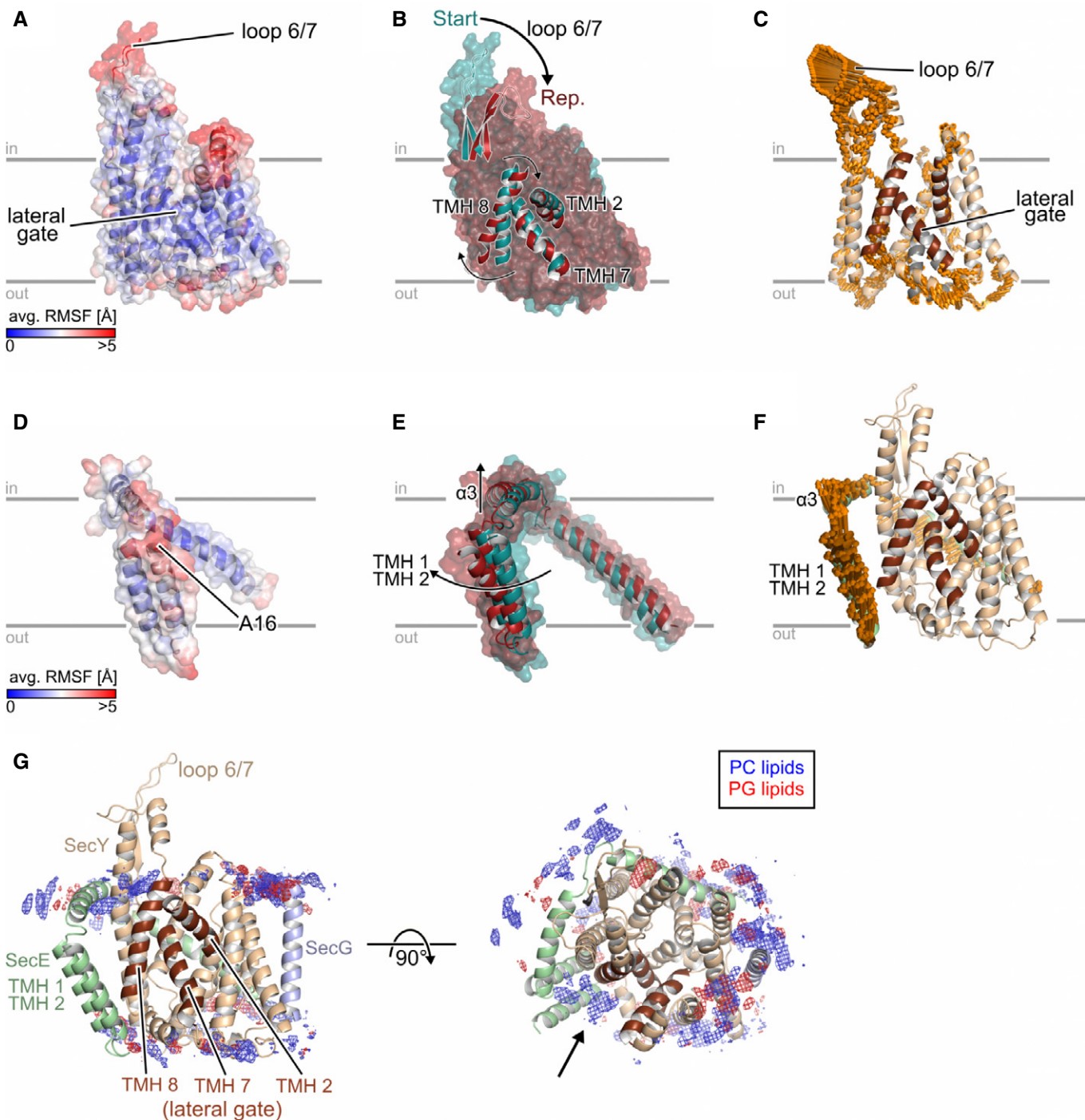

**Figure 4. Analysis of molecular dynamics simulations of SecYEG in the lipid bilayer.**

A–F (A, D): Average all-atom atomic fluctuations (RMSF; root-mean-square fluctuations) for SecY (A) and SecE (D). The mean RMSFs are projected onto the 3D structures of SecY/SecE and colored according to the color scales starting from blue (low mobility) to red (highly mobile). (B, E): Overlay of the SecY (B) or SecE (E) representative structure (red) onto the corresponding starting structure (dark cyan). Parts that show a pronounced structural change are explicitly shown as cartoon representation, and the movements are highlighted by arrows. (C, F): Visualization of displacements along the 1st (golden arrows) principal component computed for the joint, five 1 µs long MD simulations for SecY (C) and SecE (F). The amplitudes of the motions were scaled, and a cutoff for small displacements was applied for best graphical representation.

G Distribution of lipids during MD simulations. Grids represent the 3D density of phosphatidylcholine (PC; blue) and phosphatidylglycerol (PG; red) groups from the lipid bilayer. The densities were normalized to the number of considered conformations, which are identical in both cases.

reversed in the absence of the RNC, as both the distance between TMHs 3 and 7 and the angle $\eta$ increased (Fig EV4, panels B and D). Compared to the initial conformation, the distances between TMHs 2b and 7, and between TMHs 2 and 8, decreased over the course of the simulations, while the distance between TMHs 7 and 8 increased, which led to a closing of the observed V-shaped crevice (Fig EV4, panel B). Interestingly, the PC analysis also suggested that the movements of TMHs 7 and 8 were connected to the dynamics of the cytoplasmic loop 6/7 (Fig 4C, Appendix Fig S3B), such that the ribosome binding likely also influences the structural dynamics within the lateral gate, in agreement with an earlier structure of the ribosome-bound Sec61 translocon [11] and the recent biochemical data [42]. In the absence of a ribosome, binding of a short signal peptide causes an outward displacement of TMH 2b but not TMH 7 [4], so the enhanced structural dynamics at the cytoplasmic side of the lateral gate likely allows a range of pre-opened translocon conformations.

Differently to SecY, the SecE conformation from the SecYEG-ND:RNC complex corresponded to a low free energy region ($\Delta G_{conf.,i} \approx$ 0.16 kcal/mol; Appendix Fig S8A), indicating that it was similar to predominant SecE conformations in MD simulations. Accordingly, structural differences upon reaching the free energy minimum were small, as all residues in SecE, except the termini, show a RMSF < 3 Å (Fig 4D). Notably though, a small upward motion of TMHs 1 and 2 (Fig 4E and F, and Appendix Fig S4B) caused a loss of initial contacts between SecE TMH 1 and the SecY periplasmic helix (Appendix Fig S5, panel A), while new ionic interactions were formed on the periplasmic side between R44 on SecE TMH 2 and D393 on SecY TMH 9 (Appendix Fig S5, panel B). This change in the interaction pattern supports the hypothesis that the RNC-induced structural re-arrangement in SecE can be transferred toward the periplasmic part of TMHs 7, 8, and 9 in SecY and further modulates the lateral gate dynamics.

The MD simulations also revealed that the lateral gate area was enriched with zwitterionic lipids (POPC) (Fig 4G). Assuming that the phosphatidylcholine lipids used in the simulations adequately resemble the distribution of naturally occurring phosphatidylethanolamine lipids, this uneven distribution suggests that anionic lipids (POPG) are not an essential factor in the lateral gate dynamics, while the overall neutral charge in the lipid head group region may be beneficial for the insertion of hydrophobic nascent chains. The simulations furthermore indicated that anionic POPG lipids were also unevenly distributed within the nanodisc and preferentially clustered proximate to TMHs 3 and 4 of SecY (Fig 4G). Remarkably, the same regions of SecYEG have been recently described to recruit negatively charged cardiolipin lipids via interactions with lysine residues at the cytoplasmic interface of SecY, such as those in positions 115 (TMH 3) and 181 (TMH 4) [43]. Our data suggest that the SecYEG:lipid interaction is purely charge-determined, and the functionality of the translocon can be ensured either by cardiolipin or by phosphatidylglycerol lipids, while cardiolipin is not essential for the translocon functioning *in vivo* and *in vitro* [29].

As RNC FtsQ contained a hydrophobic anchor domain, we focused on locating that domain within the SecYEG-ND:RNC complex. The nascent chain could be traced along the whole ribosomal tunnel, and it was followed by a free-standing density aligned with the tunnel exit and the central cavity of SecY (Figs 5A and EV5, panel A), suggesting that the nascent chain was loaded into

the translocon. The pronounced density of TMH 2b and 7 displayed the V-shaped conformation of the partially opened lateral gate, with an additional connecting density possibly indicating the presence of a flexible or partially folded FtsQ TMH in proximity of the gate in the bilayer (Fig 5B). Finally, a short rod-like density within the nanodisc interior pointed toward the lateral gate (Fig EV5, panel A), thus suggesting that the short FtsQ TMH emerged into the bilayer via the lateral gate and acquired a stable helical conformation. As the resolution of the map alone was insufficient for unambiguous attribution of the flexible FtsQ TMH, we performed site-specific chemical cross-linking of the nascent chain and the translocon lateral gate. For this purpose, RNC variants which contained single cysteines at positions 40–43 within the FtsQ anchor domain, thus covering one helical turn, were examined. Complementary, cysteines were introduced within TMH 2b (residues 83 and 87) and TMH 7 (residues 282 and 283) of the SecY lateral gate (Fig 5B), and the cross-linking was catalyzed by copper phenanthroline. In the presence of SecY$^{C282}$EG-ND or SecY$^{C283}$EG-ND, a cross-linking product of ~80 kDa was detected for RNC FtsQ$^{C40}$ in Western blots when using antibodies against the hemagglutinin-tagged nascent chain (Fig 5C). As the molecular weight matched closely that of the putative tRNA-FtsQ:SecY adduct, we further investigated the involvement of SecY in the cross-linking products. We have found that the solvent-exposed cysteine within the periplasmic loop 3/4 of SecY (residue 148; Fig EV5, panel B) could be efficiently conjugated to CF488A-maleimide [29], but the fluorophore could not access cysteines within the lateral gate (Fig EV5, panel C). Thus, double-cysteine translocon SecY$^{C148/C283}$EG could be fluorescently labeled and used for cross-linking experiments, and presence of SecY in cross-linking adducts could be determined by in-gel fluorescence. If no ribosomes were added, only weak cross-linking products of SecY were observed at ~85 kDa that likely represented occasional translocon dimers co-reconstituted into a single nanodisc (Fig 5D). If either non-translating ribosomes or cysteine-free RNC FtsQ were added, three cross-linking bands at molecular weights between 40 and 60 kDa were observed. Those bands diminished if the sample was treated with N-ethylmaleimide prior adding the SDS-containing sample buffer (Figs 5E and EV5, panel E), so they were assigned to cross-linking of SDS-denatured SecY with ribosomal proteins. However, in the presence of RNC FtsQ$^{C40}$, a specific cross-linking product of 80 kDa was formed that agreed with the observation from Western blotting experiments (Fig 5D). Thus, we concluded that the FtsQ nascent chain indeed resided within the lateral gate and could reach the core of the translocon, but did not partition the bilayer via the cytoplasmic crevice. Interestingly, we also observed cross-linking products between SecY$^{C283}$EG-ND and the nascent chains that contained cysteines in proximate positions 41 and 42, but not the upstream position 35 (Fig 5E), thus suggesting that the N-terminal part of FtsQ TMH has been released into the lipid bilayer. The SecYEG:FtsQ cross-linking was equally efficient in the presence and absence of phosphatidylethanolamine (POPE), a major component of the bacterial membrane (Fig 5D). PE lipids are known to stimulate the SecA-mediated post-translational translocation through SecYEG [44], but seemingly have little effect on the SecYEG-ND:RNC assembly, and the translocon:ribosome complex was also visualized by cryo-EM, although at substantially lower resolution (Appendix Fig S6).

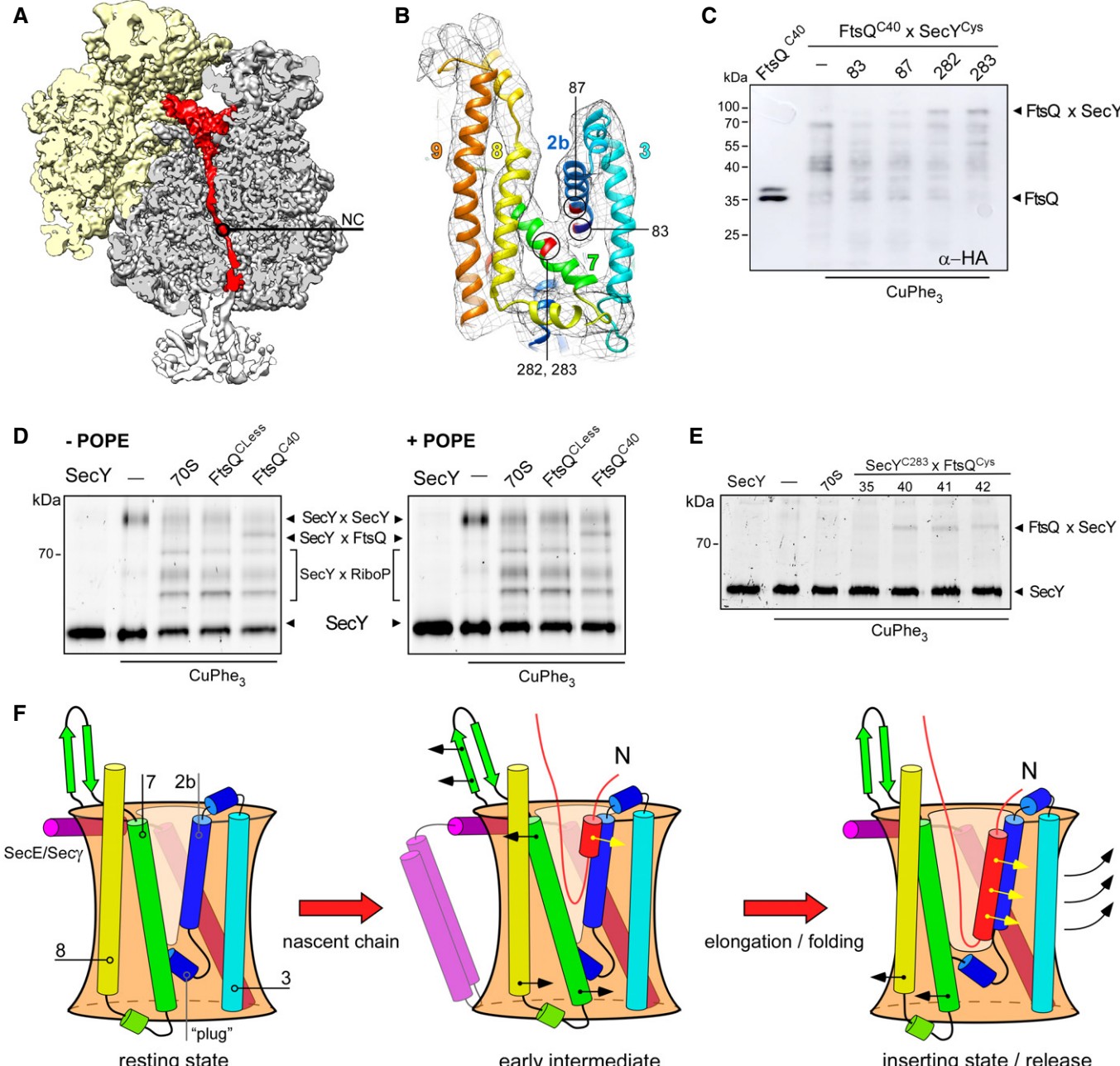

**Figure 5.  FtsQ nascent chain occupies the lateral gate of the translocon.**

A, B   Cryo-EM visualizes tRNA-bound nascent chain (red, NC) within the ribosomal tunnel (A) and a density between separated TMHs 2b and 7 of the lateral gate (B, shown in mesh). The proximate cysteine positions within SecY used for cross-linking are indicated and highlighted in red.

C   Western blot against the hemagglutinin tag within the nascent chain FtsQ$^{C40}$ reveals a cross-linking product of ~80 kDa in presence of nanodisc-reconstituted SecY$^{C282}$EG and SecY$^{C283}$EG, presumably assigned to FtsQxSecY adduct. The double band for FtsQ$^{C40}$ likely indicates the incomplete cleavage of the hexa-histidine tag by 3C protease.

D   In-gel fluorescence imaging of SecY$^{CF488A}$ reveals a cross-linking product of ~80 kDa assigned to the FtsQ$^{C40}$xSecY$^{C283}$ adduct ("SecYxFtsQ"). Bands for covalently cross-linked SecY dimer ("SecYxSecY"), and adducts of SecY and ribosomal proteins ("SecYxRiboP") are indicated. The nascent chain-specific adduct "SecYxFtsQ" does not depend on the presence of PE lipids.

E   FtsQ:SecY cross-linking is position specific. FtsQ TMH C-terminal residues 40-42 could be cross-linked to the lateral gate (SecY residue 283), while no adduct could be detected at the upstream position 35.

F   A refined scheme of the nascent chain insertion via the translocon. In the early intermediate state, the ribosome binding to the loop 6/7 and the emerging nascent chain cause displacements of SecY TMHs 2b and 7 at the cytoplasmic interface and "unzip" the lateral gate. The rigid-body tilt of TMH 7 leads to a close contact between TMHs 3 and 7 at the periplasmic side, so the V-shaped crevice is formed. The flexible nascent chain reaches the core of the lateral gate and gradually egresses the lipid moiety, where the helical fold is stabilized by hydrophobic interactions. Upon further nascent chain elongation and movements of SecY TMHs 2b, 3, 7, and 8, the widely open "inserting" state of the translocon is formed, with the complete nascent TMH at the lateral gate prior the release into the membrane.

Membrane protein biogenesis occurs in a highly complex and anisotropic environment of a lipid bilayer, and lipid:protein interactions are known to mediate the structure and functionality of inserted proteins [45–47]. Here, we have revealed the most complete structure of the lipid-embedded SecYEG translocon in complex with RNC at the early stage of the nascent chain insertion. The accessory TMHs of SecE were found to interact with the lateral gate, so they potentially mediate the gate dynamics, but may also be involved in nascent chain release or interactions with the YidC insertase [48,49] and the membrane-anchored chaperones PpiD and YfgM [50,51]. Supported by the MD simulations, the structure evidenced that the opening of the translocon lateral gate was induced at the cytoplasmic interface upon the RNC binding. TMH 2b underwent a displacement of up to ~5 Å toward the central pore, and TMH 7 tilted toward TMH 8 at the cytoplasmic side that resulted in a V-shaped crevice open for the nascent chain loading (Fig 5F). When compared to other visualized translocon:ribosome complexes, the observed translocon conformation could be readily placed between the "primed" and "inserting" states reported for the eukaryotic Sec61 complex [11,13]. Differently to the complex in its "inserting" state, the translated and exposed part of the FtsQ nascent chain was not sufficiently long to form a TMH in an $N^{in}$-$C^{out}$ topology. The cross-linking results and the weak densities observed in the cryo-EM map imply that at this early insertion stage the short nascent chain remains flexible within the lateral gate of the translocon. One can envision that the elongation of the nascent chain would cause a further displacement of SecY TMHs 2b and 7 and results in the open "inserting" state of the lateral gate, so complete folding and insertion of the TMH can be achieved in a downstream event [13]. As a bimodal profile has been observed when studying the translocon-mediated insertion of TMHs in $N^{out}$-$C^{in}$ topology, and two distinct insertion steps have been detected *in vivo* [52,53], the presented intermediate state of the translocon that allows for partial membrane partitioning and folding of a nascent TMH may potentially explain the experimental data.

# Materials and Methods

### Materials

All chemicals used were purchased from Merck/Sigma-Aldrich and Carl Roth in p.a. grade quality. Detergents were purchased from Anatrace and lipids from Otto Nordwald GmbH/Avanti Polar Lipids, Inc. Fluorophores were purchased from Thermo Fisher Scientific, Lumiprobe GmbH, and Atto-Tec GmbH.

### SecYEG purification and labeling and reconstitution

*Escherichia coli* SecYEG translocons containing an N-terminal deca-histidine tag followed by a flexible linker and the 3C protease cleavage site were overexpressed in *E. coli* strain ER2566 (New England Biolabs) and isolated as previously described [25] with minor modifications. Briefly, after the lysis (Microfluidizer M-110P, Microfluidics Corp.) bacterial membranes were pelleted upon centrifugation for 1 h at 125.000 *g* (rotor Ti45, Beckman Coulter) and resuspended in 50 mM HEPES pH 7.4, 150 mM KCl. Membranes were solubilized with 1% DDM in presence of 500 mM KCl, 50 mM HEPES pH 7.4,

200 µM TCEP, and protease inhibitors (cOmplete Protease inhibitor cocktail, Roche). Histidine-tagged translocons were isolated on $Ni^{2+}$-NTA-sepharose resin (Macherey-Nagel GmbH) following standard procedures. Optionally, labeling with 200 µM fluorophore-maleimide conjugates was carried out for 2 h prior eluting the protein from the $Ni^{2+}$-NTA resin, and the labeling efficiency was determined spectrophotometrically [29]. After the elution in presence of 300 mM imidazole, the buffer was exchanged for 50 mM HEPES pH 7.4, 150 mM KCl, 0.1% DDM, and 5% glycerol using PD SpinTrap or MiniTrap G-25 columns (GE Healthcare Life Sciences). The homogeneity of the purified translocons was controlled by size-exclusion chromatography using Superdex 200 10/300 column connected to the AKTA Purifier (GE Healthcare Life Sciences) and the protein concentrations were determined spectrophotometrically. Samples were aliquoted, flash-frozen in liquid nitrogen, and stored at −80°C.

Reconstitution of SecYEG into nanodiscs using MSP1E3D1 scaffold protein and either 30 mol % POPG, 70 mol % POPC or 30 mol % POPG, 30 mol % POPE, and 40 mol % POPC was carried out at the SecYEG:MSP:lipid molar ratio 1:10:500, as previously described [25]. SecYEG-loaded and empty nanodiscs could be separated via size-exclusion chromatography using Superdex 200 10/300 GL column.

### RNC preparation

To form SecYEG-ND:ribosome complexes, *in vivo* translation-stalled RNCs bearing FtsQ nascent chains were prepared, as previously described [15,54]. The nascent chain consisted of an N-terminal histidine tag (eight residues) linked to a 3C protease cleavage site (17 residues), FtsQ residues 4–51, hemagglutinin tag (11 residues), and the TnaC stalling motif (23 residues) (Fig EV1). For site-specific cross-linking experiments, single cysteines were introduced within the FtsQ TMH via mutations L35C, V40C, S41C, G42C, and W43C via Quick-change PCR and the mutations were confirmed by sequencing (Eurofins Genomics). The stalled nascent chain was detected via Western blot using monoclonal antibodies against the hemagglutinin tag (SC-7392) and polyclonal HRP-conjugated antibodies (SC-2005, both Santa Cruz Biotechnology).

### Cryo-EM experiments

For cryo-EM experiments, the SecYEG-ND-enriched fractions from size-exclusion chromatography were concentrated to ~1 µM by using Amicon Ultra 0.5-ml tubes (MWCO 30 kDa, Merck/Millipore), and 100 nM RNC FtsQ was added and incubated at least 15 min at room temperature. Prior to sample vitrification, fluorinated octyl-maltoside was added to the reaction to the concentration 0.2% to promote random orientation of particles on cryo-EM grids [54,55]. Vitrification was achieved using a Vitrobot mark IV (FEI). For each grid, 3.5 µl of sample was applied onto a glow discharged (20 s, 0.22 Torr) Quantifoil holey carbon grid coated with 2 nm carbon (R 3/3). After 45-s incubation, surplus sample was blotted away (2 s) and the grid was plunged into liquid ethane. From these grids, two separate datasets with a total of 13,098 micrograph movies with each 16 frames and an exposure of 2.5 $e^-$/Å²/frame were collected on a Titan Krios 300 keV cryo-electron microscope (FEI) using a Falcon II direct electron detector and the EM-Tools software (TVIPS GmbH). Magnification was set to result in a pixel size of 1,084 Å.

## Cryo-EM data analysis

Anisotropic motion correction of the micrographs was performed using MotionCor2 [56], initially using the first ten frames only. The contrast transfer function (CTF) parameters were estimated using Gctf v1.06 [57], and particles were picked using Gautomatch v0.53 (www.mrc-lmb.cam.ac.uk/kzhang/Gautomatch/). All subsequent data analysis was carried out in Relion 2.1 [58]. At first, both datasets were processed individually but identically. Two rounds of unsupervised 2D classification of all particles were performed to eliminate false positives of the particle picking step (Fig EV2). In the following step, a 3D refinement was performed to align all particles to a *E. coli* 70S ribosome reference without the translocon. All following 3D classifications were performed with fixed alignment parameters. An initial round of 3D classification with five classes was used to select for 70S particles bearing the SecYEG translocon. The resulting particles of both data sets were joined for further processing with Relion 3.0 [59]. After a further 3D refinement of the joined set, beam tilt and per particle CTF refinement was performed. Using the resulting improved CTF parameters, all particles were re-extracted with 2× binning. Multi-body refinement was used to refine the ribosomal small subunit (SSU) and ribosomal large subunit including the SecYEG-ND (LSU:SecYEG-ND) as two independent rigid bodies. Following this step, the relion_flex_analyse tool was used to subtract the signal of the SSU from the particle images and re-center these on the LSU:SecYEG-ND moiety. This process of multi-body refinement and extraction of sub-particles was then repeated for the LSU and SecYEG-ND to finally obtain a stack of particle images containing only SecYEG-ND signal. These final sub-particles were used for a further round of 3D classification. Refinement of the final subset of SecYEG-ND sub-particles resulted in an average resolution of 6.0 Å. To obtain high-resolution reconstructions of the ribosomal density, the particles of the final class were re-extracted from the motion-corrected micrographs and subjected to un-binned refinement. Again using multi-body refinement, the SSU and LSU:SecYEG-ND moieties were refined as independent rigid bodies to obtain optimal reconstructions of the ribosome, yielding resolutions of 3.3 and 3.1 Å for SSU and LSU:SecYEG:ND, respectively.

## Model building

As a starting model, we used both a crystal structure of quiescent *Thermus thermophilus* SecYEG solved in the lipid cubic phase (PDB ID: 5AWW) [4], as well as the cryo-EM structure of *Escherichia coli* SecYEG together with the 70S ribosome (PDB ID: 5GAE) [18]. Rigid-body docking was performed with UCSF Chimera [60], and the positions of individual helices were adjusted using coot [61]. To obtain reasonable geometry, real space model refinement was performed using the phenix suite [62]. To complement the intermediary resolution of the SecYEG map, the aforementioned models 5AWW and 5GAE were used to provide external reference restraints for refinement. In a final step, side chains were pruned to alanine length. Mutual orientation of SecE TMHs 1 and 2 was derived from a co-evolution pattern of residues within TMHs (http://gremlin.bakerlab.org/ecoli.php?uni = P0AG96). Strong correlations (probability score threshold 0.8) were found for residue pairs: A24:I50, L25:A54, L25:

V58, V28:L51, and A29:V48, which formed a defined interaction interface.

## Molecular dynamics simulations

In order to investigate the structural dynamics of the SecYEG complex in the absence of the ribosome and the nascent peptide, MD simulations of the SecYEG complex in an explicit membrane and explicit solvent were carried out, which used the cryo-EM-based structure as a starting conformation. ACE and NME groups were connected to the N-terminal and C-terminal residues, respectively, to avoid artificially charged termini. The SecYEG complex was prepared for pH 7 using EpiK [63] distributed with Schroedingers Maestro® suite of programs [64], which led to deprotonated residues E176 and E389 in SecY, and a protonated K81 in SecE. Furthermore, H99 in SecY was assigned to the HIE state, while the remaining histidine residues are in the HID state. We used the in-house software packmol_memgen, now also distributed with the Amber 18 suite of programs [65], to embed the SecYEG complex into a POPC:POPG (ratio 2:1) bilayer that mimics the nanodisc composition, to add 0.15 M of KCl, and to solvate the bilayer system with TIP3P water [66]. All relevant system files for subsequent MD simulations were generated using the LeaP program of the Amber 17 suite of programs [67]. The Amber ff14SB force field [68] was used to parametrize the protein, adaptations by Joung and Cheatham [69] were applied to treat $K^+$ and $Cl^-$, and the lipid 17 force field distributed with Amber 17 to treat the lipid bilayer.

For subsequent MD simulations, we used the simulation protocol as described by us previously [70,71]. In order to set up five independent MD production simulations, the target temperature during thermalization varied from 299.8 to 300.2 K in 0.1 K intervals, so that we obtained five different configurations for subsequent MD production runs. These production simulations were performed at 300.0 K for 1.0 µs. Coordinates were saved in a trajectory file every 200 ps. The particle mesh Ewald method was applied to treat long-range electrostatic interactions. Structural relaxation, thermalization, and production runs of MD simulations were conducted with pmemd.cuda [72] of Amber 17 [67].

We used the cpptraj program [73] to analyze the trajectories with respect to distances, root-mean-square fluctuations (RMSF), a measure for atomic mobility, angles, and lipid distributions. If not reported differently, all results are expressed as mean value ± standard error of mean over $n = 5$ independent simulations. Additionally, we performed a principal component analysis to extract the essential motions displayed by the systems, after superimposing each snapshot onto the ten transmembrane helices in SecY of the overall average coordinates in order to remove global rotational and translational motions. Mapping SecY and SecE along the trajectories onto a plane spanned by the 1st and 2nd principal components yielded a 2D histogram, from which we estimated the relative configurational free energy $\Delta G_{conf.,i}$ of the state of the protein in bin $i$ using equation (1)

$$\Delta G_{conf.,i} = -RT \ln \frac{N_i}{N_{\max}} \tag{1}$$

where $R$ is the universal gas constant, $T = 300$ K, $N_i$ the population of bin $i$, and $N_{\max}$ the population of the most populated bin [74].

The representative conformations of SecY and SecE were extracted from the MD trajectories and analyzed toward their structural features relative to the initial 3D structure.

### *In vitro* cross-linking

To probe potential SecYEG:FtsQ contacts, 100 nM RNC FtsQ variants bearing single cysteines (mutations V40C, S41C, G42C, and W43C) in the FtsQ TMH were incubated with ~1 μM SecYEG-ND, which contained single cysteines within the translocon lateral gate (M83C, S87C, I282C, and I283C). After 15-min incubation at the ambient temperature, cupper phenanthroline was added to the concentration of 1 mM, and the cross-linking reaction was conducted for 30 min at the ambient temperature. Cross-linking products containing the nascent chain were detected via Western blotting [75]. Western blots were developed using ECL Western blotting substrate (Pierce) and imaged using LAS-4000 Mini imager (GE Life Sciences). To detect SecYEG-based cross-linking products, the cysteines within the lateral gate were combined with a cysteine at the translocon periplasmic interface (mutation L148C), which was labeled with CF488A-maleimide (Sigma/Merck), as previously described [29]. For the cross-linking experiments, 100 nM SecY$^{CF488A}$EG-ND variants was mixed with 200 nM non-translating ribosomes or RNCs, and the cross-linking with copper phenanthroline was conducted as described above. Where indicated, samples were treated with *N*-ethylmaleimide prior loading on SDS–PAGE. In-gel fluorescence was recorded using Typhoon FLA 7000 imaging system (GE Life Sciences).

## Data availability

The datasets produced in this study are available in the following databases:

- cryo-EM map: Electron Microscopy Data Bank (EMDB, www.ebi.ac.uk/pdbe/emdb), accession code 4743.
- SecYEG molecular model: Protein Data Bank (PDB, www.rcsb.org), accession code 6R7L.

**Expanded View** for this article is available online.

## Acknowledgements
We would like to acknowledge Susanne Rieder for the assistance with the cryo-EM sample preparation; Bertrand Beckert and André Heuer for the support with the data processing; and Eli van der Sluis for fruitful discussions. L.K. is a scholar of the QBM Graduate School. The research was supported by the Deutsche Forschungsgemeinschaft (DFG) via the Research grant KE1879/3-1 to A.K. and projects A10 (A.K.) and A03 (H.G.) within the CRC 1208 "Identity and dynamics of biological membranes" (project number 267205415), and European Research Council (ERC, Advanced Investigator grant "Cryotranslation" to R.B.). The computational support and infrastructure for MD simulations were provided by the Centre for Information and Media Technology (ZIM) at Heinrich Heine University Düsseldorf. We are grateful to the John von Neumann Institute for Computing (NIC) and the Jülich Supercomputing Centre for computing time on the supercomputer JURECA (NIC project HKF7, H.G. and B.F.). Financial support by DFG for funds (INST 208/704-1 FUGG) to purchase the hybrid computer cluster used in this study is gratefully acknowledged by H.G.

## Author contributions
LK: cryo-EM data analysis and structure modeling. BF: molecular dynamics simulations and free energy calculations. OB: cryo-EM data acquisition and analysis. HG: data analysis of molecular dynamics simulations, project design, and supervision. RB: data analysis, project design, and supervision. AK: biochemical preparations and experiments, cryo-EM data analysis, project design, and supervision. All authors contributed to the manuscript preparation.

## Conflict of interest

The authors declare that they have no conflict of interest.

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
