## [Review Process File · EMBO Reports]

Partially inserted nascent chain unzips the lateral gate of the Sec translocon

Lukas Kater, Benedikt Frieg, Otto Berninghausen, Holger Gohlke, Roland Beckmann, Alexej Kedrov

Review timeline:

Submission date:	29 March 2019
Editorial Decision:	9 May 2019
Revision received:	5 June 2019
Editorial Decision:	4 July 2019
Revision received:	10 July 2019
Accepted:	16 July 2019

Transaction Report:

1st Editorial Decision

29 March 2019

Thank you for the submission of your research manuscript to our journal. We have now received the full set of referee reports that is copied below.

As you will see, the referees acknowledge the interest of your findings and support publication in EMBO reports after relatively minor revisions.

Given these constructive and supportive comments, we would like to invite you to revise your manuscript with the understanding that the referee concerns (as detailed above and in their reports) must be fully addressed and their suggestions taken on board. Please address all referee concerns in a complete point-by-point response. Acceptance of the manuscript will depend on a positive outcome of a second round of review. It is EMBO reports policy to allow a single round of revision only and acceptance or rejection of the manuscript will therefore depend on the completeness of your responses included in the next, final version of the manuscript.

Revised manuscripts should be submitted within three months of a request for revision; they will otherwise be treated as new submissions. Please contact us if a 3-months time frame is not sufficient for the revisions so that we can discuss the revisions further.

I include below some important points to consider before submitting your revised manuscript:

Please list the accession numbers and database for the cryo-EM and molecular model in a formal "Data Availability" section (placed after Materials & Method) that follows the model below (see also <http://embor.embopress.org/authorguide#dataavailability>). Please note that the Data Availability Section is restricted to new primary data that are part of this study.

Data availability

- [data type]: [name of the resource] [accession number/identifier/doi] ([URL or identifiers.org/DATABASE:ACCESSION])

Our journal also encourages inclusion of *data citations in the reference list* to directly cite datasets that were re-used and obtained from public databases. Data citations in the article text are distinct from normal bibliographical citations and should directly link to the database records from which the data can be accessed. In the main text, data citations are formatted as follows: "Data ref: Smith et al, 2001" or "Data ref: NCBI Sequence Read Archive PRJNA342805, 2017". In the Reference list, data citations must be labeled with "[DATASET]". A data reference must provide the database name, accession number/identifiers and a resolvable link to the landing page from which the data can be accessed at the end of the reference. Further instructions are available at < <http://embor.embopress.org/authorguide#referencesformat>>.

Supplementary/additional data: The Expanded View format, which will be displayed in the main HTML of the paper in a collapsible format, has replaced the Supplementary information. You can submit up to 5 images as Expanded View. Please follow the nomenclature Figure EV1, Figure EV2 etc. The figure legend for these should be included in the main manuscript document file in a section called Expanded View Figure Legends after the main Figure Legends section. Additional Supplementary material should be supplied as a single pdf labeled Appendix. The Appendix includes a table of content on the first page with page numbers, all figures and their legends. Please follow the nomenclature Appendix Figure Sx throughout the text and also label the figures according to this nomenclature. For more details please refer to our guide to authors.

Regarding data quantification, please ensure to specify the name of the statistical test used to generate error bars and P values, the number (n) of independent experiments underlying each data point (not replicate measures of one sample), and the test used to calculate p-values in each figure legend. Discussion of statistical methodology can be reported in the materials and methods section, but figure legends should contain a basic description of n, P and the test applied. Please also include scale bars in all microscopy images.

We now strongly encourage the publication of original source data with the aim of making primary data more accessible and transparent to the reader. The source data will be published in a separate source data file online along with the accepted manuscript and will be linked to the relevant figure. If you would like to use this opportunity, please submit the source data (for example scans of entire gels or blots, data points of graphs in an excel sheet, additional images, etc.) of your key experiments together with the revised manuscript. Please include size markers for scans of entire gels, label the scans with figure and panel number, and send one PDF file per figure.

- a complete author checklist, which you can download from our author guidelines (<http://embor.embopress.org/authorguide#revision>). Please insert page numbers in the checklist to indicate where the requested information can be found.
 - a letter detailing your responses to the referee comments in Word format (.doc)
 - a Microsoft Word file (.doc) of the revised manuscript text
 - editable TIFF or EPS-formatted figure files in high resolution
- (In order to avoid delays later in the publication process please check our figure guidelines before preparing the figures for your manuscript:
http://www.embopress.org/sites/default/files/EMBOPress_Figure_Guidelines_061115.pdf)
- a separate PDF file of any Supplementary information (in its final format)
 - all corresponding authors are required to provide an ORCID ID for their name. Please find instructions on how to link your ORCID ID to your account in our manuscript tracking system in

our Author guidelines (<http://embor.embopress.org/authorguide>).

As part of the EMBO publication's Transparent Editorial Process, EMBO reports publishes online a Review Process File to accompany accepted manuscripts. This File will be published in conjunction with your paper and will include the referee reports, your point-by-point response and all pertinent correspondence relating to the manuscript.

I look forward to seeing a revised version of your manuscript when it is ready. Please let me know if you have questions or comments regarding the revision.

REFeree REPORTS

Referee #1:

In their manuscript Kater et al. explore the structure of an early intermediate of a ribosome-nascent chain complex in contact with the SecYEG translocon. By using Cryo-EM, MD simulations and biochemical assays, they show that a short nascent chain induces a partial opening of SecY's lateral gate and that SecE unexpectedly appears to modulate lateral gate movements. The authors further suggest that the nascent chain remains in an unfolded or loosely folded state until it is exposed to the lipid environment.

Despite a wealth of structural and biochemical data, our understanding on how the SecYEG translocon mediates membrane protein insertion is still enigmatic and the structural characterization of an early insertion intermediate is therefore an important step forward in understanding this mechanism. The manuscript is well written and the data in general support the conclusions. Still, I have a few issues the authors should address for improving their study.

Major issues:

1. The authors suggest that the nascent chain achieves proper folding only when reaching the lipids. This conclusion is based on cross-linking experiments, which show that three consecutive residues (FtsQ40-FtsQ42) can be cross-linked to a single SecY residue. In my opinion, these observations are not sufficient to conclude that the nascent chain is indeed unfolded/loosely folded. Helix formation has been shown to occur already within the ribosomal tunnel and the cross-link pattern could also be explained by different orientations of a folded helix within the lateral gate. Thus the authors should tone down their statement or back it up by further experiments, e.g. FRET analyses.
2. Based on the CBB-stained gel in Fig. 1C, it appears that SecE/G are sub-stoichiometric to SecY. Although it is known that SecG is not easily stained by CBB, at least SecE should be detectable. Could the authors use a different staining method or immune detection for demonstrating that they have indeed primarily reconstituted SecYEG into their nanodiscs?

Minor issues:

1. Page 1: to me the description of SecE sounds as if SecE would have four transmembrane domains, which is not the case. The authors should rephrase this part.
2. The authors might want to refer to data by Beck et al. (2000) and Houben et al (2004). In both studies RNC cross-links to SecE have been observed, which was difficult to reconcile with the structure of the SecYEG complex. The current structure might provide an explanation for these contacts.
3. Fig. 1: The MW of SecY seems to be rather high; it usually migrates at approx. 35 kDa, but here it migrates at approx. 42 kDa. Is there any special reason?
4. In general, the manuscript would be much easier to comprehend, if the TMH and loops in the individual figures would be labeled; e.g. Fig. 3, Fig. 4G
5. The occasional SecY dimers seem to be rather pronounced in the absence of ribosomes/RNCs and

- are much weaker in their presence. Does this suggest that ribosomes/RNCs dissociate SecY dimer?
3. Add MW marker to Fig. 5E
 4. In Fig. 5F, the early intermediate state depicts the nascent chain completely within the translocon, while Fig. S1 and the text leave the impression that the N-terminus of the RNC is already outside. This should be consistent.
 5. Fig. S1: include length of the different parts of the RNC; typo: 'hemagglutinin'
 6. Fig. S4: typo 'quiescent'
 7. Fig. S5 needs labeling, define color code
 8. Fig. S10E is not described in the legend; define *
 9. Reference 51 is incomplete

Referee #2:

Membrane insertion of proteins is co-translationally achieved by the Sec translocon complex and the ribosome-nascent chain complex (RNC). Although many biochemical and biophysical analyses have been performed, the structural changes that SecYEG (bacterial Sec translocon) undergoes during this process are still not known. From the point of view of basic research, it is very important to obtain a structural snapshot of SecYEG. In this study, the most accurate structure of the SecYEG-nanodisc-RNC complex obtained thus far has been shown. SecYEG in the lipid bilayer of the nanodisc represents its pre-open form, possibly induced by the binding of the RNC complex. The EM structure shows separate densities of SecYEG TMHs, which is the most important and reliable point of this study. The conformational changes in SecYEG were supported by means of molecular dynamics simulation using the new structural model of SecYEG without the RNC complex. Importantly, I would like to point out that the reliability of the MD simulation would be low because of the moderate resolution model of SecYEG. However, performing the MD simulation as a successive analysis was a nice approach in this paper. Although it was very difficult to clearly identify the position of the substrate protein on the SecYEG from the presented EM data, the authors elucidated the positions of the substrate protein at the crevasse of SecYEG by performing site-directed crosslink analysis. As these data present key information for understanding membrane protein insertion, I strongly recommend publishing this paper in EMBO reports. To make your article easier to understand, I recommend that you address the following points:

A cited paper (Tanaka et al., 2015) in the text has presented the crystal structure (5CH4) of SecYEG with a peptide in its crevasse. In your paper, a comparison with this structure may be interesting. I suggest that the authors include and discuss this comparison.

In Fig. 1A, the SecYEG structure is from *Thermus thermophilus*, and not *Thermatoga maritima*. Please make this change.

In Fig. 1A, please label the sides, "in" and "out".

In Fig. 2B, please include the scale bar.

In Fig. 3D, please label the numbers of TMHs; this will help the readers easily understand this figure.

In Fig. 5B, the additional map does not seem to be clear. Please modify it for clarity.

In Fig. S4, please illustrate the membrane planes, and label the sides "in" and "out". Expanding the figure size would be better because the figures are a little complicated to comprehend.

In Fig. 10B, please indicate the precise mutated positions.

For Fig. 10E, the legend is missing. Please include this.

Because PE is one of the components of bacterial membrane, the EM map of SecYEG-nanodisc using POPG/POPE/POPC in Fig. S11 conveys important information. Please include the FCS graph in Fig. S11. In Fig. S11, the SecYEG-nanodisc (POPG/POPC) structure should be included to compare the structures. A superimposed figure may be apt to depict similar structures. Additionally, I recommend including detailed comments about the structure in the text.

Referee #1:

In their manuscript Kater et al. explore the structure of an early intermediate of a ribosome-nascent chain complex in contact with the SecYEG translocon. By using Cryo-EM, MD simulations and biochemical assays, they show that a short nascent chain induces a partial opening of SecY's lateral gate and that SecE unexpectedly appears to modulate lateral gate movements. The authors further suggest that the nascent chain remains in an unfolded or loosely folded state until it is exposed to the lipid environment.

Despite a wealth of structural and biochemical data, our understanding on how the SecYEG translocon mediates membrane protein insertion is still enigmatic and the structural characterization of an early insertion intermediate is therefore an important step forward in understanding this mechanism. The manuscript is well written and the data in general support the conclusions.

Authors: Thank you for your positive comments!

Still, I have a few issues the authors should address for improving their study. Major issues:

Reviewer 1, Comment 1: *The authors suggest that the nascent chain achieves proper folding only when reaching the lipids. This conclusion is based on cross-linking experiments, which show that three consecutive residues (FtsQ40-FtsQ42) can be cross-linked to a single SecY residue. In my opinion, these observations are not sufficient to conclude that the nascent chain is indeed unfolded/loosely folded. Helix formation has been shown to occur already within the ribosomal tunnel and the cross-link pattern could also be explained by different orientations of a folded helix within the lateral gate. Thus the authors should tone down their statement or back it up by further experiments, e.g. FRET analyses.*

Authors: Thank you for the suggestion! We based our model of the partially unfolded nascent chain on the combination of cross-linking results, absence of helix-like densities in the cryo-EM map, and also geometrical considerations, which largely exclude the fully folded nascent chain forming an insertion hairpin within the translocon due to the insufficient length of the polypeptide chain. However, we agree that those considerations are rather indirect and alternative interpretations of the experimental results are possible. Thus, we have modified the abstract, discussion and the legend of Fig. 5F to describe a flexible state of the partially inserted nascent chain.

Reviewer 1, Comment 2: *Based on the CBB-stained gel in Fig. 1C, it appears that SecE/G are substoichiometric to SecY. Although it is known that SecG is not easily stained by CBB, at least SecE should be detectable. Could the authors use a different staining method or immune detection for demonstrating that they have indeed primarily reconstituted SecYEG into their nanodiscs?*

Authors: A new SDS-PAGE of the SecYEG-ND preparation after size exclusion chromatography have been included in the revised manuscript that validates the presence of SecE at stoichiometric amounts to SecY. In agreement, the presented cryo-EM map reveals SecY and SecE densities at comparable signal levels, thus verifying the stable inter-subunit assembly upon the translocon reconstitution into nanodiscs.

Minor issues:

Reviewer 1, Comment 3: *Page 1: to me the description of SecE sounds as if SecE would have four transmembrane domains, which is not the case. The authors should rephrase this part.*

Authors: The text has been modified for clarity.

Reviewer 1, Comment 4: *The authors might want to refer to data by Beck et al. (2000) and Houben et al (2004). In both studies RNC cross-links to SecE have been observed, which was difficult to reconcile with the structure of the SecYEG complex. The current structure might provide an explanation for these contacts.*

Authors: Thank you for the suggestion! Indeed, co-translational cross-linking of both MtlA and the leader peptidase (Lep) to SecE have been previously described at different stages of insertion. The visualized proximity of SecE TMH 1 to the lateral gate may explain the crosslinks with the nascent chains being sufficiently long to be exposed beyond the lateral gate, as it is the case for Lep in Houben et al. (2004) *EMBO Reports*. However, clarifying the crosslinks between SecE and the nascent chain residues near the ribosomal exit tunnel as reported by Beck and co-workers is more challenging and would require structural analysis of the translocon engaged with the particular nascent chain (MtlA). The discussion and the reference to Houben et al. have been added to the manuscript (page 7).

In the revised manuscript we have also included a reference to a recent publication from Reid Gilmore's lab that describes the dynamics of the homologous Sec61 translocon and shows that the residues between the loop 6/7 and TMH 7 play a key role in opening of the lateral gate (Mandon et al. 2018 *J Biol Chem*) that is in a remarkable agreement with our results on the coupled dynamics of the ribosome-binding loop and the lateral gate.

Reviewer 1, Comment 5: *Fig. 1: The MW of SecY seems to be rather high; it usually migrates at approx. 35 kDa, but here it migrates at approx. 42 kDa. Is there any special reason?*

Authors: The increase in the apparent MW of SecY upto 40 kDa (Fig. 1C and EV5, panels C and D) is likely attributed to 28 amino acids extension at the N-terminal end: a deca-histidine tag followed by a flexible linker and the 3C protease cleavage site. The extension would not only increase the molecular weight of the protein by 3 kDa, but it also increases its polarity and interactions with SDS, thus affecting the migration within SDS-PAGE. The details on the used SecYEG variant have been added to the Methods section.

Reviewer 1, Comment 6: *In general, the manuscript would be much easier to comprehend, if the TMH and loops in the individual figures would be labeled; e.g. Fig. 3, Fig. 4G*

Authors: Following the advice (also Reviewer 2, Comment 5) labelling of the secondary structure elements was added to Figures 3 and 4.

Reviewer 1, Comment 7: *The occasional SecY dimers seem to be rather pronounced in the absence of ribosomes/RNCs and are much weaker in their presence. Does this suggest that ribosomes/RNCs dissociate SecY dimer?*

Authors: We agree with the Reviewer that there is a pronounced effect of RNCs on the appearance of SecY:SecY crosslinked adducts in Fig. 5D. Indeed, one hypothesis would be a dissociation of dimeric translocons, or a conformational change within the lateral gate that prevents SecY:SecY crosslinking. However, since the reconstituted SecYEG dimers represent a minute fraction of nanodiscs formed in excess of lipids and MSP (Taufik, Kedrov et al. 2013 *J Mol Biol*), and also a dual orientation of protomers within a disc is possible, explaining the role of the RNC binding on SecYEG quaternary dynamics based on the crosslinking results would be rather ambiguous.

Reviewer 1, Comment 8: *Add MW marker to Fig. 5E*

Authors: Position of the pre-stained 70 kDa MW marker was added to the image.

Reviewer 1, Comment 9: *In Fig. 5F, the early intermediate state depicts the nascent chain completely within the translocon, while Fig. S1 and the text leave the impression that the N-terminus of the RNC is already outside. This should be consistent.*

Authors: Thank you for the comment! As the structural and biochemical experiments suggest that the N-terminal end of the nascent chain leaves the translocon, the scheme in Fig. 5F has been modified. Additionally, the TMHs of the lateral gate have been indicated for clarity.

Reviewer 1, Comment 10: *Fig. S1: include length of the different parts of the RNC; typo: 'hemagglutinin'*

Authors: The lengths of each part has been added and the typo corrected. In the revised manuscript the figure is listed as the Expanded View Figure EV1.

Reviewer 1, Comment 11: *Fig. S4: typo 'quiescent'*

Authors: The typo has been corrected.

Reviewer 1, Comment 12: *Fig. S5 needs labeling, define color code*

Authors: Labeling and an explanation for the color code has been added. The rainbow color scale encodes $C\alpha$ -distance between the two aligned models ranging from 0 Å (blue) to 3 Å or greater (red). In the revised manuscript the figure is listed as the Expanded View Figure EV3.

Reviewer 1, Comment 13: *Fig. S10E is not described in the legend; define **

Authors: The figure panel has been described. Thank you for the remark! In the revised manuscript the figure is listed as the Expanded View Figure EV5.

Reviewer 1, Comment 14: *Reference S1 is incomplete.*

Authors: The reference has been corrected.

Referee #2:

Membrane insertion of proteins is co-translationally achieved by the Sec translocon complex and the ribosome-nascent chain complex (RNC). Although many biochemical and biophysical analyses have been performed, the structural changes that SecYEG (bacterial Sec translocon) undergoes during this process are still not known. From the point of view of basic research, it is very important to obtain a structural snapshot of SecYEG. In this study, the most accurate structure of the SecYEG-nanodisc-RNC complex obtained thus far has been shown. SecYEG in the lipid bilayer of the nanodisc represents its pre-open form, possibly induced by the binding of the RNC complex. The EM structure shows separate densities of SecYEG TMHs, which is the most important and reliable point of this study. The conformational changes in SecYEG were supported by means of molecular dynamics simulation using the new structural model of SecYEG without the RNC complex. Importantly, I would like to point out that the reliability of the MD simulation would be low because of the moderate resolution model of SecYEG. However, performing the MD simulation as a successive analysis was a nice approach in this paper. Although it was very difficult to clearly identify the position of the substrate protein on the SecYEG from the presented EM data, the authors elucidated the positions of the substrate protein at the crevasse of SecYEG by performing site-directed crosslink analysis. As these data present key information for understanding membrane protein insertion, I strongly recommend publishing this paper in EMBO reports.

Authors: Thank you for the positive evaluation!

To make your article easier to understand, I recommend that you address the following points:

Reviewer 2, Comment 1: A cited paper (Tanaka et al., 2015) in the text has presented the crystal structure (5CH4) of SecYEG with a peptide in its crevasse. In your paper, a comparison with this structure may be interesting. I suggest that the authors include and discuss this comparison.

Authors: We apologize for leaving the mentioned structure out of the initial manuscript. Indeed, the structure 5CH4 may be interpreted as a very early intermediate in the lateral gate opening by a signal sequence, where the outward displacement of TMH 2b widens the lateral gate by ~3 Å relatively to the quiescent state (5AWW). Interestingly, our molecular model, as well as previous high-resolution structures describe inwards movement of TMH 2b upon the nascent chain insertion in the presence of a ribosome. Furthermore, no movement of TMH 7 is observed in 5CH4, that agrees with our findings that the dynamics of TMH 7 is mediated by the ribosome docking on the loop 6/7. The reference and a short discussion have been added to the manuscript (page 8).

Reviewer 2, Comment 2: In Fig. 1A, the SecYEG structure is from *Thermus thermophilus*, and not *Thermatoga maritima*. Please make this change.

Authors: Thank you for the remark! We apologize for the mistake.

Reviewer 2, Comment 3: In Fig. 1A, please label the sides, "in" and "out".

Authors: The membrane interfaces in Fig. 1A and also the Appendix Figure S2 (formerly S4) have been indicated as "cytoplasm" and "periplasm".

Reviewer 2, Comment 4: In Fig. 2B, please include the scale bar.

Authors: The scale bar (20 nm) has been added.

Reviewer 2, Comment 5: In Fig. 3D, please label the numbers of TMHs; this will help the readers easily understand this figure.

Authors: Following the Reviewer's advice, labelling of the secondary structure elements was added.

Reviewer 2, Comment 6: In Fig. 5B, the additional map does not seem to be clear. Please modify it for clarity.

Authors: We have modified the figure (transparency levels, the view angle) to improve the presentation quality.

Reviewer 2, Comment 7: In Fig. S4, please illustrate the membrane planes, and label the sides "in" and "out". Expanding the figure size would be better because the figures are a little complicated to comprehend.

Authors: The approximate membrane borders and sidedness have been indicated. We also optimised the figure assembly that, we hope, allow for better view of superimpositions. In the revised manuscript the figure is listed as the Appendix Figure S2.

Reviewer 2, Comment 8: In Fig. 10B, please indicate the precise mutated positions.

Authors: Cysteine positions within the lateral gate have been specified according to the Reviewer's suggestion.

Reviewer 2, Comment 9: For Fig. 10E, the legend is missing. Please include this.

Authors: The figure panel has been described. Thank you for the remark!

Reviewer 2, Comment 10: Because PE is one of the components of bacterial membrane, the EM map of SecYEG-nanodisc using POPG/POPE/POPC in Fig. S11 conveys important information. Please include the FCS graph in Fig. S11. In Fig. S11, the SecYEG-nanodisc (POPG/POPC) structure should be included to compare the structures. A superimposed figure may be apt to depict similar structures. Additionally, I recommend including detailed comments about the structure in the text.

Authors: The presented cryo-EM map serves as a direct proof of the RNC:translocon complex assembly in presence of PE lipids, and an overlay of SecYEG-ND cryo-EM density map with the built molecular model has been included in the revised manuscript, as it is suggested by the Reviewer (the figure is currently listed as the Appendix Figure S6). Unfortunately, the cryo-EM reconstruction obtained in presence of POPE did not allow for detailed structural analysis due to the limited resolution (filtered at 9 Å). Upon single-particle analysis we have observed high heterogeneity within SecYEG-ND, that agrees well with previous structural and biochemical studies on PE-containing nanodiscs (Shenkarev *et al.* 2013, *BBA*; Henrich *et al.* 2017 *eLife*). One possible explanation is poor compatibility of PE lipids with planar bilayers, as it has been documented for DOPE, but may also account for POPE employed in our work. Furthermore, the relatively high transition temperature of POPE (25 °C vs. 3 °C for POPG and POPC) may cause separation of lipids based of physico-chemical properties under ambient temperature, thus adding to the complexity and the heterogeneity of the nanodisc preparation.

2nd Editorial Decision

5 June 2019

Thank you for your patience while we have editorially assessed your revised manuscript. I am now writing with an 'accept in principle' decision, which means that I will be happy to accept your manuscript for publication once a few minor issues/corrections have been addressed, as follows.

- Please provide a legend for movie EV1 as follows: the legend is a simple README.txt file. Then the legend and the movie are zipped together and the zip file is uploaded (i.e., one .zip file including legend and movie).

- Please provide a link to the database in the Data availability section. The description of the data(base) should follow this model:

- [data type]: [name of the resource] [accession number/identifier/doi] ([URL or identifiers.org/DATABASE:ACCESSION])

- Please provide up to five keywords on the first page of the manuscript.

- Figure callouts in the text: We noticed that the callouts to Appendix Figs S1 and S2 are missing the word 'Appendix'.

- In our routine analysis of all figures we noticed the following issues that need your attention:

- Fig 2A has a blank square in the top left corner. Please remove it.

- Fig 5C,D, and E appear overcontrasted. Please supply scans with less contrast modification, if available.

- Fig 4C and Appendix Fig S3B are identical. Please clearly indicate this in the figure legend. The same applies to Fig 4F and Appendix Fig S4B.

- Our data editors from Wiley already inspected the Figure legends for completeness and accuracy. Please see their suggested changes in the attached Word file. I have also taken the liberty to make a small change to the title. Could you please review it?

Once you have made these minor revisions, please use the following link to submit your corrected manuscript:

<https://embor.msubmit.net/cgi-bin/main.plex?el=A5Ij5GF3A1CqJZ6J1A9ftdsu50j9SHFRh8lAFkutt22QY>

If all remaining corrections have been attended to, you will then receive an official decision letter from the journal accepting your manuscript for publication in the next available issue of EMBO reports. This letter will also include details of the further steps you need to take for the prompt inclusion of your manuscript in our next available issue.

Thank you for your contribution to EMBO reports.

2nd Revision - authors' response

10 July 2019

The authors performed all minor editorial changes.

Corresponding Author Name: Dr. Alexej Kedrov, Prof. Dr. Roland Beckmann, Prof. Dr. Holger Gohlke

Manuscript Number: EMBOR-2019-48191